# PKCδ deficiency inhibits fetal development and is associated with heart elastic fiber hyperplasia and lung inflammation in adult PKCδ knockout mice

Yuko S. Niino[1¤a], Ikuo Kawashima[2], Yoshinobu Iguchi[3], Hiroaki Kanda[4], Kiyoshi Ogura[2], Kaoru Mita-Yoshida[5], Tomio Ono[5], Maya Yamazaki[6¤b], Kenji Sakimura[6¤c], Satomi Yogosawa[7], Kiyotsugu Yoshida[7], Seiji Shioda[1¤d], Takaya Gotoh[8]*

1 Department of Anatomy, Showa University of Medicine, Shinagawa-ku, Tokyo, Japan, 2 Laboratory of Biomembrane, Tokyo Metropolitan Institute of Medical Science, Setagaya-ku, Tokyo, Japan, 3 Technology Research Division, Tokyo Metropolitan Institute of Medical Science, Setagayaku, Tokyo, Japan, 4 Department of Pathology, Saitama Cancer Center, Saitama, Kita-adachi-gun, Saitama, Japan, 5 Laboratory for Transgenic Technology, Tokyo metropolitan Institute of Medical Science, Setagayaku, Tokyo, Japan, 6 Department of Cellular Neurobiology, Brain Research Institute, Niigata University, Niigata, Japan, 7 Department of Biochemistry, The Jikei University School of Medicine, Minato-ku, Tokyo, Japan, 8 Faculty of Sports and Health Science, Department of Health Science, Daito Bunka University, Higashimatsuyama, Saitama, Japan

¤a Current address: Faculty of Sports and Health Science, Department of Health Science, Daito Bunka University, Higashimatsuyama, Saitama, Japan
¤b Current address: Department of Neurology, University of California, San Francisco, California, United State of America
¤c Current address: Department of Animal Model Development, Brain Research Institute, Niigata University, Niigata, Japan
¤d Current address: Department of Clinical Pharmacy, Shonan University of Medical Sciences, Totsuka-ku, Yokohama, Japan
* gotoh_tky@ic.daito.ac.jp

**Data Availability Statement:** All relevant data are within the paper and its Supporting information files.

## Abstract

Protein kinase C-delta (PKCδ) has a caspase-3 recognition sequence in its structure, suggesting its involvement in apoptosis. In addition, PKCδ was recently reported to function as an anti-cancer factor. The generation of a PKCδ knockout mouse model indicated that PKCδ plays a role in B cell homeostasis. However, the *Pkcrd* gene, which is regulated through complex transcription, produces multiple proteins via alternative splicing. Since gene mutations can result in the loss of function of molecular species required for each tissue, in the present study, conditional PKCδ knockout mice lacking PKCδI, II, IV, V, VI, and VII were generated to enable tissue-specific deletion of PKCδ using a suitable Cre mouse. We generated PKCδ-null mice that lacked whole-body expression of PKCδ. PKCδ+/- parental mice gave birth to only 3.4% PKCδ-/- offsprings that deviated significantly from the expected Mendelian ratio ($\chi2(2) = 101.7$, $P < 0.001$). Examination of mice on embryonic day 11.5 (E11.5) showed the proportion of PKCδ-/- mice implanted in the uterus in accordance with Mendelian rules; however, approximately 70% of the fetuses did not survive at E11.5. PKCδ-/- mice that survived until adulthood showed enlarged spleens, with some having cardiac and pulmonary abnormalities. Our findings suggest that the lack of PKCδ may have harmful effects on fetal development, and heart and lung functions after birth. Furthermore,

**Funding:** This work was supported by the JSPS KAKENHI Grant-in-Aid for Scientific Research (C) grant numbers 21590204:YSN and 15K10689: YSN, and JSPS KAKENHI Grant Number JP 16H06276 (AdAMS):HK. The funders had no role in study design, data collection and analysis, decision to publish, or preparation of the manuscript.

**Competing interests:** The authors have declared that no competing interests exist.

our study provides a reference for future studies on PKCδ deficient mice that would elucidate the effects of the multiple protein variants in mice and decipher the roles of PKCδ in various diseases.

## Introduction

Protein kinase C (PKC) is a phospholipid-dependent serine/threonine kinase, first identified in 1977 by Nishizuka et al., and it plays critical roles in intracellular signal transduction [1–3]. Mammalian PKCs form a large family, categorized based on their molecular structures and activation mechanisms as conventional PKC (cPKC), requiring calcium, phosphatidylserine, and diacylglycerol for activation (α, β, and γ isoforms); novel PKC (nPKC), which do not require calcium for activation (δ, ε, η, and θ isoforms); and atypical PKC, which do not require calcium or diacylglycerol for activation (ζ and λ/i isoforms). PKCδ belongs to the nPKC family, and PKCδI was the first PKCδ molecular species to be reported. PKCδI is expressed ubiquitously in various tissues and cells, suggesting a general role rather than a tissue- or cell-specific function [4]. PKCδ is involved in many cellular processes [5,6], including cell growth [7], apoptosis [8,9], tumor inhibition [10], and cell migration [11]. cPKCs and nPKCs, including PKCδ, are activated by the oncogenic promoter phorbol 12-myristate 13-acetate and, hence, are considered as drivers of tumorigenesis [4]. However, despite over 30 years of clinical trials investigating PKC inhibitors as anti-cancer agents, PKC inhibitors have failed to show tumor-suppressive effects and, in some cases, have worsened symptoms [12]. Detailed studies have identified PKCδ as a tumor suppressor [13]. Additionally, PKCδ has a caspase-3 cleavage sequence, called the DILD motif, in its V3 domain, indicating a role in apoptosis [6,7,14–16].

Studies based on PKCδ knockout mice (PKCδ KO) have demonstrated that PKCδ is involved in the maintenance of smooth muscle homeostasis and that PKCδ KO mice develop normally and are fertile [17]. PKCδ plays a critical role in B cell homeostasis and tolerance, highlighting its potential role in the treatment of autoimmune diseases [18,19]. Although these studies reported significant findings, the PKCδ KO mice used in these studies lacked only the PKCδI and δII isoforms. Studies have shown that several PKCδ isoforms are generated by the alternative splicing from single PKCδ gene (*Prkcd*), and that PKCδ itself forms a family (Fig 1). To date, the expression levels of PKCδI, II, IV, V, VI, VII, and IX have been reported in mice [20–22], those of PKCδI and III in rats [23], and those of PKCδI and VIII in humans [24]. In mice, caspase-3 recognition sequences are present in PKCδI, IV, and VI. PKCδII, V, and VII have a 78-bp insertion sequence with no frameshift in the DILD motif; therefore, PKCδII, V, and VII are not cleaved by caspase-3 [21]. PKCδVIII, the human homolog of the mouse PKCδII [24], exerts anti-apoptotic effects in NT2 cells [25,26]. Moreover, insulin enhances cell growth by further promoting alternative splicing of PKCδII in HT22 cells [27]. It has been suggested that the PKCδ variants may have distinct functions.

Exons are indicated by orange, red, light blue, and yellow boxes and numbered according to a report on the mouse genome by Suh et al. [28]. Red, exon 4a; blue, 78 bp insertion in the caspase-3 recognition site; yellow, specific exon for PKCδVI and VII; M, the first methionine; and asterisks, the termination codon. Introns are shown with horizontal lines. PKCδ isoform-characteristic exons are shown under the schematic structures of the entire PKCδ genomic DNA.

The complex regulation of gene transcription and the production of multiple splice variants, each with their own functions, indicates the efficiency of gene usage. However, mutations

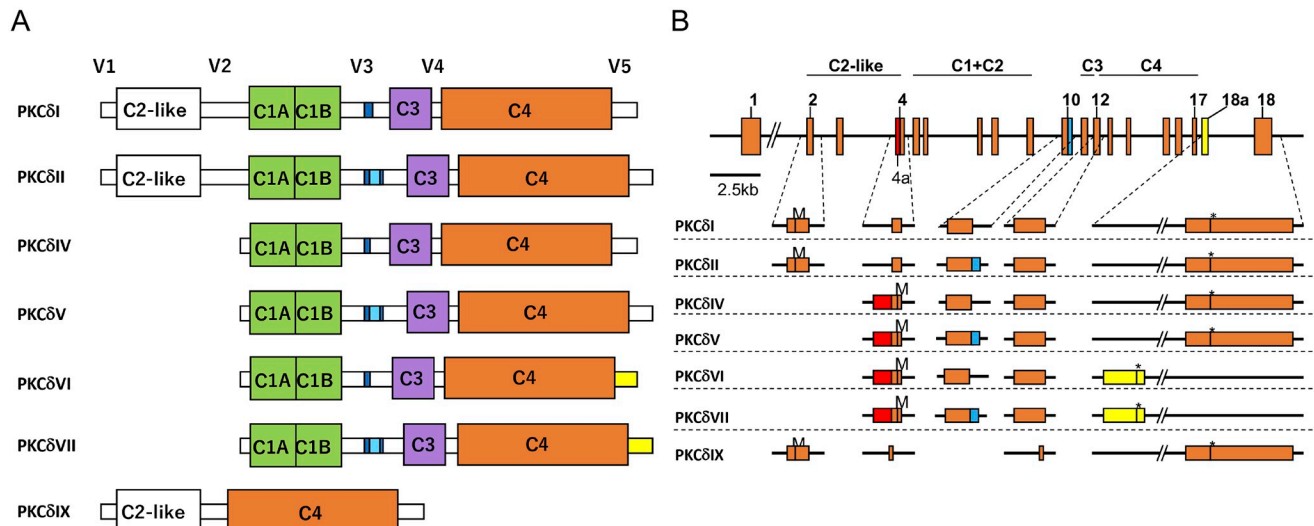

**Fig 1. Schematic structures of mouse protein kinase C delta (PKCδ) isoforms and genomic DNA.** (A) PKCδI and II cDNAs consist of variable domains V1, V2, V3, V4, and V5, and conserved domains C2-like, C1a, C1b, C3, and C4. The C1 regulatory domain contains two cysteine-rich zinc finger motifs. PKCδIV, V, VI, and VII consist of C1a, C1b, V3, C3/V4/C4, and V5 domains. Dark blue, DILD caspase-3 recognition site; light blue, 78 bp insertion in DILD caspase-3 recognition site; and yellow, specific exon for PKCδVI and VII [20–22]. (B) Schematic structure of the PKCδ genome.

at key points in a gene can result in the simultaneous loss of multiple proteins. In the present study, we generated mice with comprehensive deletion of the PKCδ-encoding gene, to elucidate the functions of PKCδ. Thus, in this study, a new PKCδ-deficient mouse model was generated by deleting the exon shared by PKCδI, II, IV, V, VI, and VII. We report that this mouse model yielded different data than what has been previously described.

## Materials and methods

### Mouse breeding

All mice procedures were carried out in accordance with the guidelines laid down by the Institutional Animal Care and Use Committees and the Ethics Committees of Tokyo Metropolitan Institute of Medical Science (approval number 15085, 16018, 17026, 18010), Niigata University (approval number SA00542), and Showa University (approval number 50040, 51014, 52016) approved this study.

### Generation of PKCδ-flox mice

Three genomic DNA fragments of the PKCδ gene (*Prkcd*): 4.6 kb of the 5′ arm, 0.85 kb of the region including exon 7, and 6 kb of the 3′ arm were amplified using polymerase chain reaction (PCR) from the mouse genomic bacterial artificial chromosome RP23-283B12 (Thermo Fisher Scientific, Waltham, Massachusetts, USA). The targeting vector contained the 5′ arm gene fragment upstream of the first loxP sequence, pgk-1 promoter-driven neomycin phosphotransferase gene, and exon 7 (86 bp) of the PKCδ gene downstream of the first loxP sequence, with the 3′ arm gene fragment downstream of the second loxP sequence. Additionally, the targeting vector contained a diphtheria toxin gene for negative selection (Fig 2A), and was electroporated into the C57BL/6N-derived embryonic stem (ES) cell line, RENKA [29], after linearization by SalI digestion. Homologous recombinants were identified by Southern blotting. After digestion with EcoRI and BamHI, the genomic DNA samples were hybridized

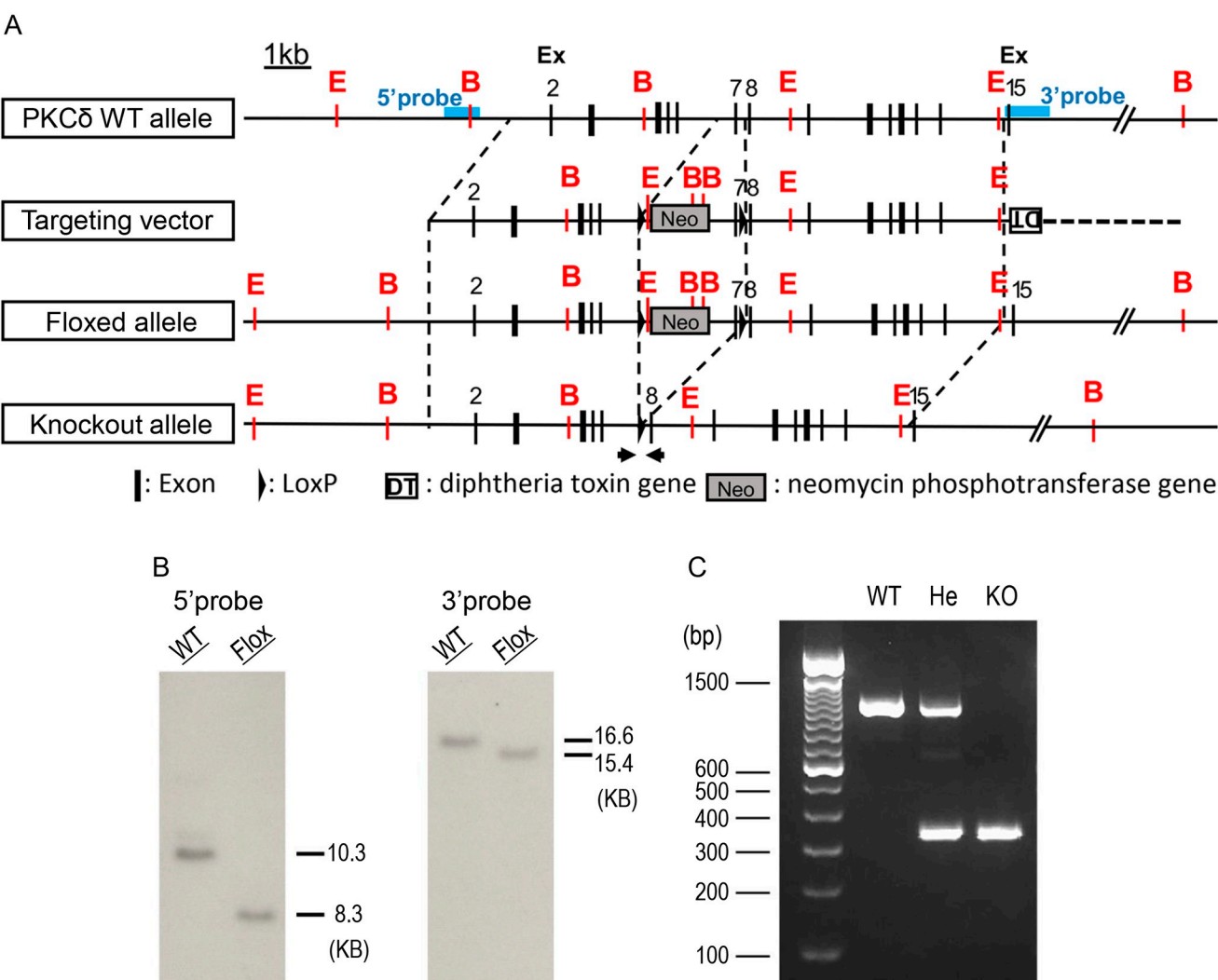

**Fig 2. Generation of protein kinase C delta (PKCδ)-deficient mice.** (A) Schematic representation of murine *Prkcd* genes, targeting vectors, targeted allele, and knockout allele. Numbers denote the exon numbers, and light blue boxes denote 5′- and 3′- probes for Southern blotting. Polymerase chain reaction (PCR) primers for genotyping are indicated by arrows. Red letters E and B represent EcoRI and BamHI, respectively. Dashed line indicates vector sequence. (B) Southern blotting using genomic DNA of wild type (WT) and flox+/flox+ mouse. DNA digested using EcoRI was used for 5′-probes and shows sizes of 10.3 KB for WT and 8.3 kb for flox+/flox+ mice. DNA digested using BamHI was used for 3′-probes and shows sizes of 16.6 kb for WT and 15.4 kb for flox +/flox+ mice. (C) Genotypic analysis by PCR. PCR analysis was performed using genomic DNA extracted from 4-week-old mice tails and the amniotic membranes of fetuses. WT shows bands of 1078 bp, KO of 315 bp, and He of both 1078 and 315 bp.

with 5′- and 3′-probes, as shown in Fig 2A. DNA was separated on a 0.6% agarose gel, followed by transfer to a Biodyne Plus membrane (Pall Corp., New York, USA). The membrane was hybridized with DNA probes labeled with the PCR DIG probe synthesis kit (Roche, Basel, Switzerland). The probes were detected with an alkaline phosphatase-conjugated anti-DIG antibody and visualized with a DIG luminescent detection kit (Roche). EcoRI digested genomic DNA hybridized with a 5′-probe showed 10.3 kb for wild type (WT) and 8.2 kb for the targeted allele. BamHI digested genomic DNA hybridized with the 3′-probe showed 16.6 kb for the WT allele and 15.4 kb for the targeted allele (Fig 2B). ES cells with the correct recombination were used to produce chimeric mice.

## Generation of PKCδ deficient mice

PGK2-Cre mice [Tg (Pgk2-cre) 24Shb] [30] maintained at Showa University, specifically express Cre recombinase in spermatocytes, and were used for mating with the PKCδ-flox mice to obtain PKCδ-deficient mice. From the offsprings, we selected PKCδ+/- (He, hetero-type knockout) and Cre-/- mice and mated them to obtain homozygous PKCδ-/- mice (KO, PKCδ knockout).

## Genotyping PKCδ deficient mice using PCR

Genotyping was performed by PCR using genomic DNA from the tail of 4-week-old off-springs, and the amnion from the embryos using the Tks Gflex DNA polymerase (TAKARA, Kusatsu, Shiga, Japan) under the following conditions: 2 min at 94˚C, 35 cycles of 20 sec at 94˚C, 1 min at 68˚C, followed by 2 min at 68˚C. The primers used for genotyping are given in Fig 2, and their sequences are as follows: forward, 5′-GCAGGTGGTGAGTGTTCCTT-3′; reverse, 5′-GGCATGTCGATGTTGAAGCG-3′. The size of the PCR products was 1078 bp for WT and 315 bp for KO allele.

## Histological analysis

Hearts, lungs, and spleens were removed from WT (PKCδ+/+) and KO mice at 16 and 24 weeks of age for histological analysis. After mating 11 pairs of PKCδ He mice, E11.5 embryos were collected after confirming the number of embryo sacs. Genomic DNA from the amniotic membranes from E11.5 was used for fetal genotyping. Murine tissues and E11.5 embryos were fixed using 4% paraformaldehyde solution at 4˚C overnight, dehydrated, defatted through an ethanol- and xylene-series, and then paraffin-embedded. Each tissue was sliced into 5-μm thick sections and stained with Hematoxylin and Eosin (H&E stain). The heart and lung sections were stained using Masson's trichrome (MT) and Elastica van Gieson staining (EVG) to stain collagen fibers and elastic fibers, respectively. We used Hematoxylin solution made by Muto Pure Chemicals Co. Ltd. (Tokyo) and Eosin solution by Sakura Finetek Japan Co. Ltd. (Tokyo). For both MT and EVG, we used staining solutions made by Muto Pure Chemicals Co. Ltd. (Tokyo).

## Statistical analysis

The statistical difference between our results and Mendel's law was analyzed using the chi-square test. The Kaplan–Meier method was used to analyze the survival of PKCδ KO mice.

# Results

## Generation of PKCδ-deficient mice

In the present study, PKCδ-deficient mice were generated to analyze the function of PKCδ. PKCδ is reported to generate eight molecular species (PKCδI, II, IV, V, VI, VII, and IX) from a single gene (*Prkcd*) in mice [20–22]. The structures of the mouse PKCδ molecular species in mice are shown in Fig 1A. Studies on PKCδ knockout mice (KO) reported that KO mice were not deficient in all PKCδ molecular species [17,18]. Therefore, to fix the lineage of PKCδ knockout mice into C57BL6/N mice, we generated conditional knockout C57BL6/N mice with deficient PKCδI, II, IV, V, VI, and VII using the RENKA, C57BL/6N ES cell line.

The relationship between the PKCδ gene structure and the different molecular species generated by alternative splicing is shown in Fig 1B, with the genetic architecture previously described by Suh et al. [28]. In the present study, we inserted LoxP sequences in the upstream and downstream regions of exon 7 of 86 base pairs, that could reliably frameshift due to the

defect and prevent protein generation (Fig 2A). Homologous recombination was performed using the RENKA C57BL/6N ES cell line [29] to generate PKCδ conditional knockout mice (PKCδ flox/+). PKCδ flox/flox mice were further generated by self-mating PKCδ flox/+ mice, which were then enrolled (registered: C57BL/6 Prkcd<tm1Shb>). Confirmation of this genetic insertion of PKCδflox/ flox mice was performed by genomic Southern blotting with C57BL/6N wild-type mice (Fig 2B). PKCδ flox mice were crossed with PGK2–Cre mice [Tg (Pgk2-cre) 24Shb] [30] to generate conditional knockout of PKCδ genes (PKCδ KO). Since PGK2–Cre mice express Cre recombinase in spermatocytes and spermatids in the testis, the resulting male mice have a PKCδ gene deletion in their spermatozoa. PKCδ+/- mice with Cre-/- genotypes obtained by crossing with wild-type PKCδ+/+ (WT) were enrolled as PKCδ +/-, which means PKCδ gene Heterozygous mice (He) (registered: C57BL/6-Prkcd<tm1. 1Shb>).

Male and female PKCδ He mice were mated, offspring were weaned four weeks after delivery, and the genotypes were confirmed using PCR. Representative PCR data are shown in Fig 2C. The genotypes of the resulting offspring at weaning are indicated in Table 1. The proportion of genotypes in the offspring mice was expected to be WT:He:KO = 1:2:1, whereas the observed proportion was WT:He:KO = 104 (27.1%):267 (69.5%):13 (3.4%). The genotypes of all 384 offspring born from pairs of PKCδ He mice did not follow Mendelian rules (By $\chi^2$ test, $\chi^2 = 101.7$, p< 0.001).

The lifespan of all PKCδ KO mice born to pairs of PKCδ He mice was as follows: 2.5 months (n = 1), 3 months (n = 2), 6 months (n = 1), 7 months (n = 1), 11 months (n = 2), 13 months (n = 1), and 15 months (n = 1). Four other knockout mice were used in the experiment. These nine PKCδ KO mice were kept simultaneously with their littermate WT and He mice, and their Kaplan-Meier survival curves are shown in Fig 3. However, since WT and He mice were euthanized at 1 year (52 weeks old), the study by Yuan R. et al. [31] was cited as supplementary data for WT. No spontaneous death was observed in WT and He mice within 1 year. In contrast, KO mice started to die when they were 10 weeks old. As a result, it was found that the 50% survival rate of the KO mice was 31.8 weeks, which was 1/4 that of WT mice.

## Histological analysis

Eight mice were euthanized for histological analysis. Four PKCδ KO mice (one male and two female 24-week-old mice, one male 16-week-old mouse), and four gender- and age-matched WT mice descend from a pair of PKCδ He parents. Visual observations revealed that all PKCδ KO mice had enlarged spleens compared to those of PKCδ WT mice, and the hearts were enlarged in three of the four PKCδ KO mice (Fig 4A). Additionally, conspicuous deformation was observed in the lungs of three out of four KO mice (Fig 4A). These findings are summarized in Table 2. No visible differences were observed in the other organs. Histological analysis

**Table 1. Genotypic analysis of 4-week-old offspring mice at weaning from heterozygous parents.**

|  | WT | He | KO |
|---|---|---|---|
| Male | 53 | 144 | 3 |
| Female | 51 | 123 | 10 |
| Total | 104 | 267 | 13 |
| Rate of offspring | 27.1% | 69.5% | 3.4% |

The genotypes of all 384 offspring born from pairs of PKCδ. He mice did not follow Mendelian rules ($\chi^2$(2) = 101.7, p < 0.001).

Genotypes: WT, wild type mice; He, hetero-type knockout mice; KO, homozygous PKCδ-/- mice (PKCδ knockout).

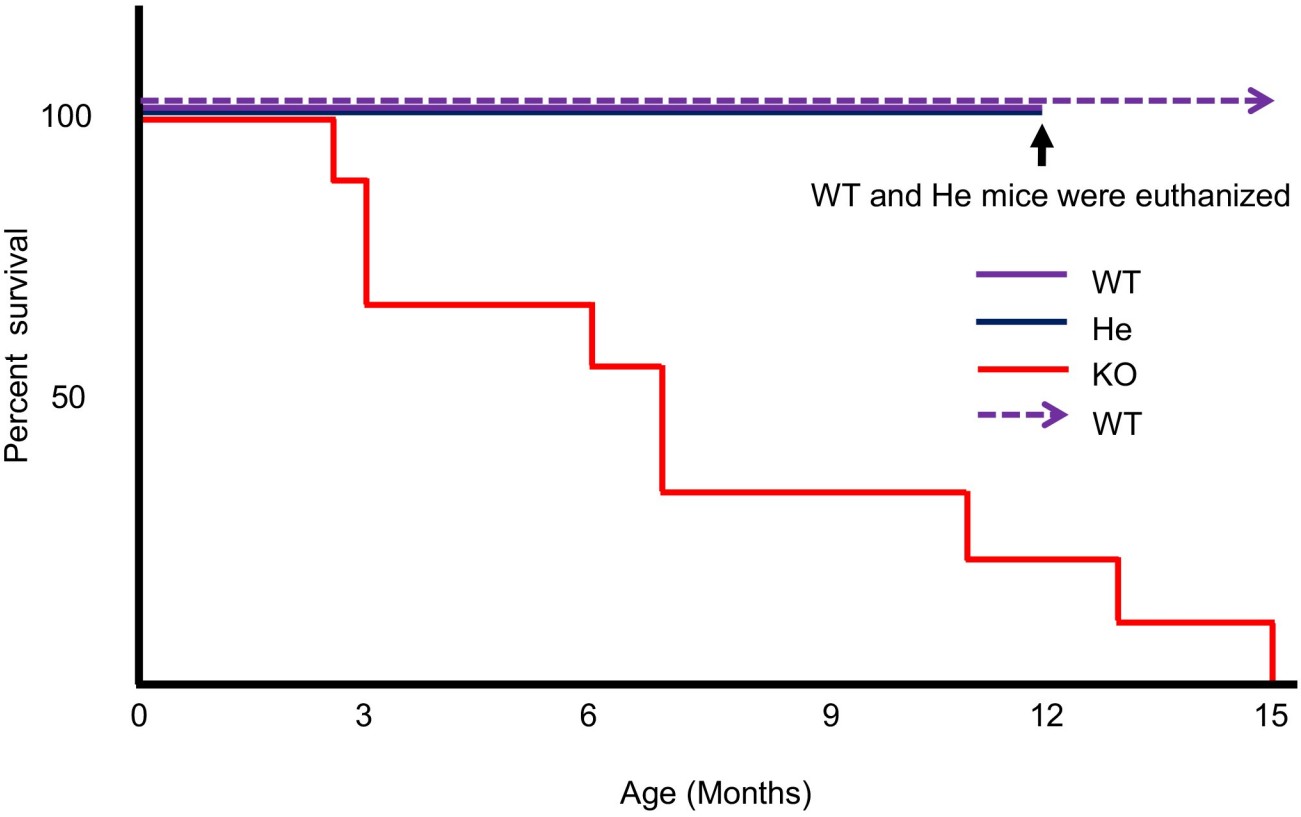

**Fig 3. Survival curve of PKCδ KO mice (Kaplan-Meier plots).** PKCδ He mice were spontaneously bred, the genotypes of the offsprings were determined, and their survival was monitored. KO mice (all nine KO mice) were observed until all mice died, whereas WT and He mice were euthanized at 52 weeks of age. Therefore, the study by Yuan et al. [31] is cited as supplementary data for WT and is indicated by the dashed arrow.

was performed for each of the above organs. Calcification was observed in the heart of two out of four PKCδ KO mice, and inflammation was observed in the lungs of three out of four KO mice, indicating poorly air space, as observed by H&E staining (Fig 4B). In the two cases with a cardiac abnormality, a calcific mass above the left ventricle in one mouse (Fig 4B and 4C) and calcification was observed at the base of the cardiac mitral valve in the other mouse (Fig 4D) were observed. The calcification in both cases observed in the hearts of male mice. MT and EVG staining were performed on heart sections of mice with calcification in the heart to examine the reason for the calcification. In MT staining, collagen fibers were stained blue with aniline blue solution and the cytoplasm was stained red with Masson's solution. In EVG staining, elastic fibers were stained black with resorcinol fuchsin solution, collagen fibers were stained red with Sirius red solution, and muscle fibers were stained yellow with picric acid. The staining indicated a clear increase in elastic fibers in the hearts of PKCδ KO mice compared to the hearts of WT mice (Fig 5A). In both PKCδ KO mice hearts, EVG staining revealed that there was a hyperplasia of the elastic fibers in the endocardium. In addition, there was a slight trend in the blood vessels of the heart. Furthermore, the results of MT staining showed an increase in collagen fibers; however, the changes in elastic fibers were more intense and predominant. This may have resulted in lime-induced thrombus; however, a causal relationship could not be established. An increase and calcification of elastic fibers in the endocardium were observed in the hearts of two of the four PKCδ KO mice. However, it was difficult to determine whether this change was significant. In addition, MT and EVG staining were

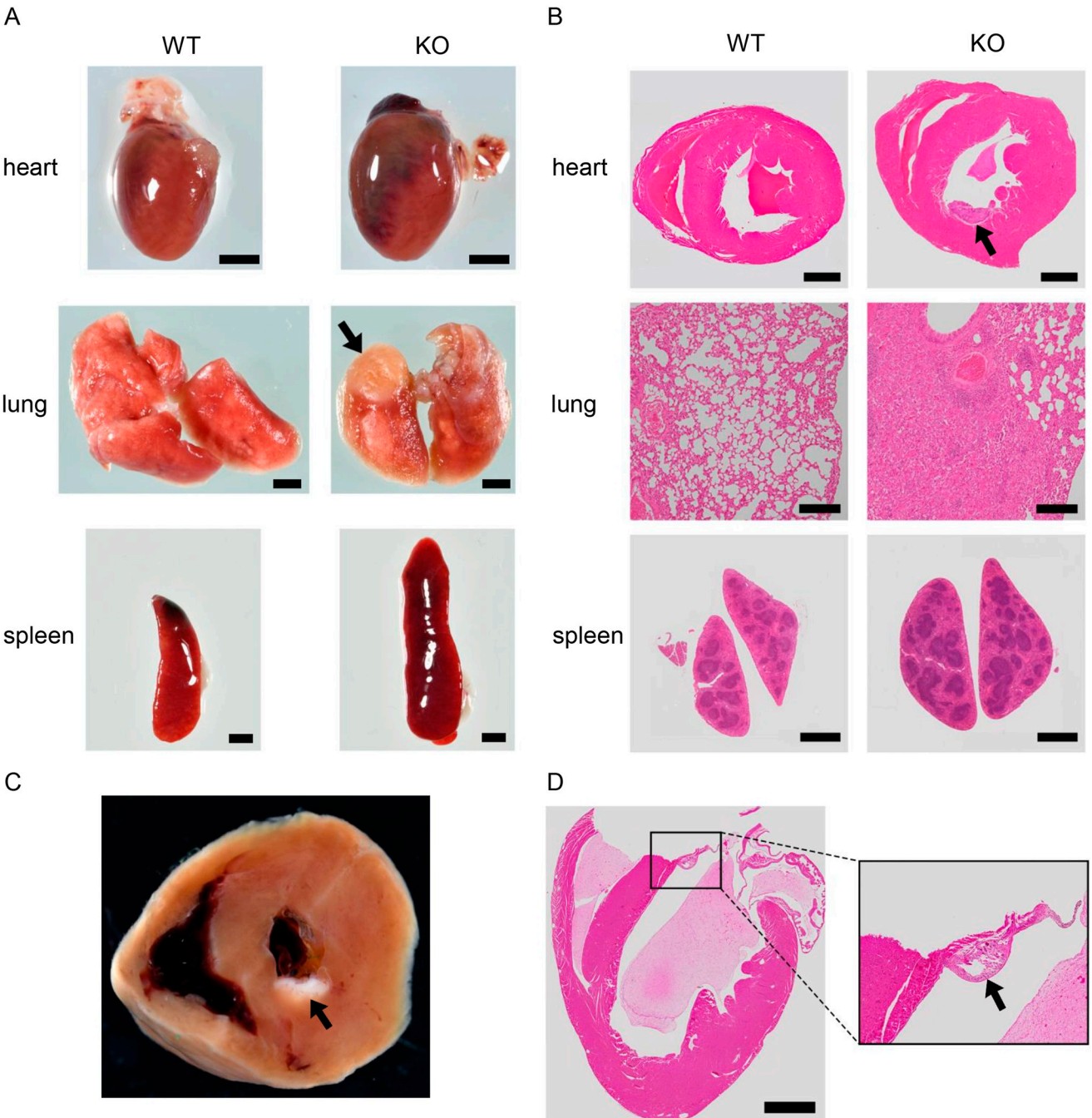

**Fig 4. Histological analysis of WT and KO mice.** (A) Comparison of heart and spleen sizes between WT and KO mice. Arrow indicates the site of deformity in the lung. Scale bars represent 2 mm. (B) Hematoxylin and eosin-stained images of hearts, lungs, and spleens from WT and KO mice. The heart image shows a cross-section. Scale bars represent 1 mm for the heart and spleen and 200 μm for the lung. Arrow indicates the site of calcification. (C) Heart of a KO mouse with internal calcification. Arrow indicates the site of calcification. (D) Hematoxylin and eosin-stained images of vertically cut hearts from KO mice. Inset shows calcification at the base of the mitral valve. Scale bars represent 1 mm. Arrow indicates the site of calcification.

**Table 2. Phenotypic analysis of KO-mice.**

|  | Age | Sex | Heart | | Lung | Spleen |
|---|---|---|---|---|---|---|
|  |  |  | Enlarged | Calcification | Inflammation | Enlarged |
| 1 | 16 w | Male | + | + | + | + |
| 2 | 24 w | Male | + | + | + | + |
| 3 | 24 w | Female | + | - | + | + |
| 4 | 24 w | Female | - | - | - | + |

Age, sex, and phenotype of the mice (n = 4) subjected to histological analysis are shown.

performed on the hearts of PKCδ KO mice without cardiac enlargement and there was no difference was observed compared to the WT mice. Examination of lung abnormalities revealed that three PKCδ KO animals showed more eosinophilic exudates and macrophages in the alveoli, and more round cell infiltrates compared with those of the PKCδ WT mice. Macrophages were also found in the bronchi. Despite strong congestion and inflammation in the lungs, the increase in fiber was inconspicuous. Lung abnormalities appeared to be a change resulting from decreased cardiac function (Fig 5B).

## Analysis of 11.5-day-old embryos

To verify the reason for the low production rate of approximately 3.4% of PKCδ KO mice from crosses between PKCδ He mice, we analyzed fetuses on embryonic day 11.5 (E11.5). PKCδ He mice were self-crossed, and 11 pregnant mice were examined. The uterus was removed, and the fetus was collected after confirming the number of fetal sacs. The total number of fetal sacs was 104, and 103 fetuses were identified. One fetal sac contained no fetus, as it had already been resorbed. The fetal implantation rate of each genotype is shown in Table 3A. Genotypes were determined by PCR using genomic DNA from the amniotic membrane. The PCR patterns of WT, He, and KO mice are shown in Fig 2C. Since the PKCδIV, V, VI, and VII isoforms were expressed in a testis-specific manner, we thought that PKCδ knockout mice might have some problems in the early stages of development such as fertilization, early cleavage, and implantation. However, our results demonstrated the presence of 8–12 fetal sacs in the uterus of 10 out of 11 mother mice. The genotype of the fetuses in the fetal sacs at E11.5 is shown in Table 3A. It suggests that all genotypes of WT, He, and KO follow the Mendelian rule ($\chi 2(2)$ = 2.077, n.s.), and fertilization, early cleavage, and implantation (placenta confirmed) were not problematic. However, as shown in Table 3B, the survival rate of KO fetuses was significantly lower than that of the WT and He embryos at E11.5 ($\chi 2(2)$ = 30.892, p <0.001). This suggests that KO fetuses have problems developing in the mother's body, which explains the one possible reason for the low birth rate of KO mice. The implantation rate of PKCδ KO fetuses was 23.1% of the total, but the survival rate decreased in the mother's body and further decreased to 3.4% at 4 weeks of age (Table 1).

As representative E11.5 samples, fetuses of mother numbers 8 and 11 are shown in Fig 6A and 6B. Fetuses with heartbeats on E11.5 (fetus numbers 2 (WT) and 7 (KO) in Fig 6A, and 3 (WT) and 10 (KO) in Fig 6B) were fixed using 4% paraformaldehyde solution and stained with HE for histological examination (Fig 7A and 7B). The fetuses of PKCδ KO mice were slightly smaller than those of WT mice, and their Sclerotome and spinal cord appeared to be immature compared with those of the WT mice (Figs 6 and 7). Although there was no difference in the size of the heart between KO and WT mice, the ventricular wall was thin in the KO mice. These are especially mesoderm-derived organs. Additionally, there were no obvious

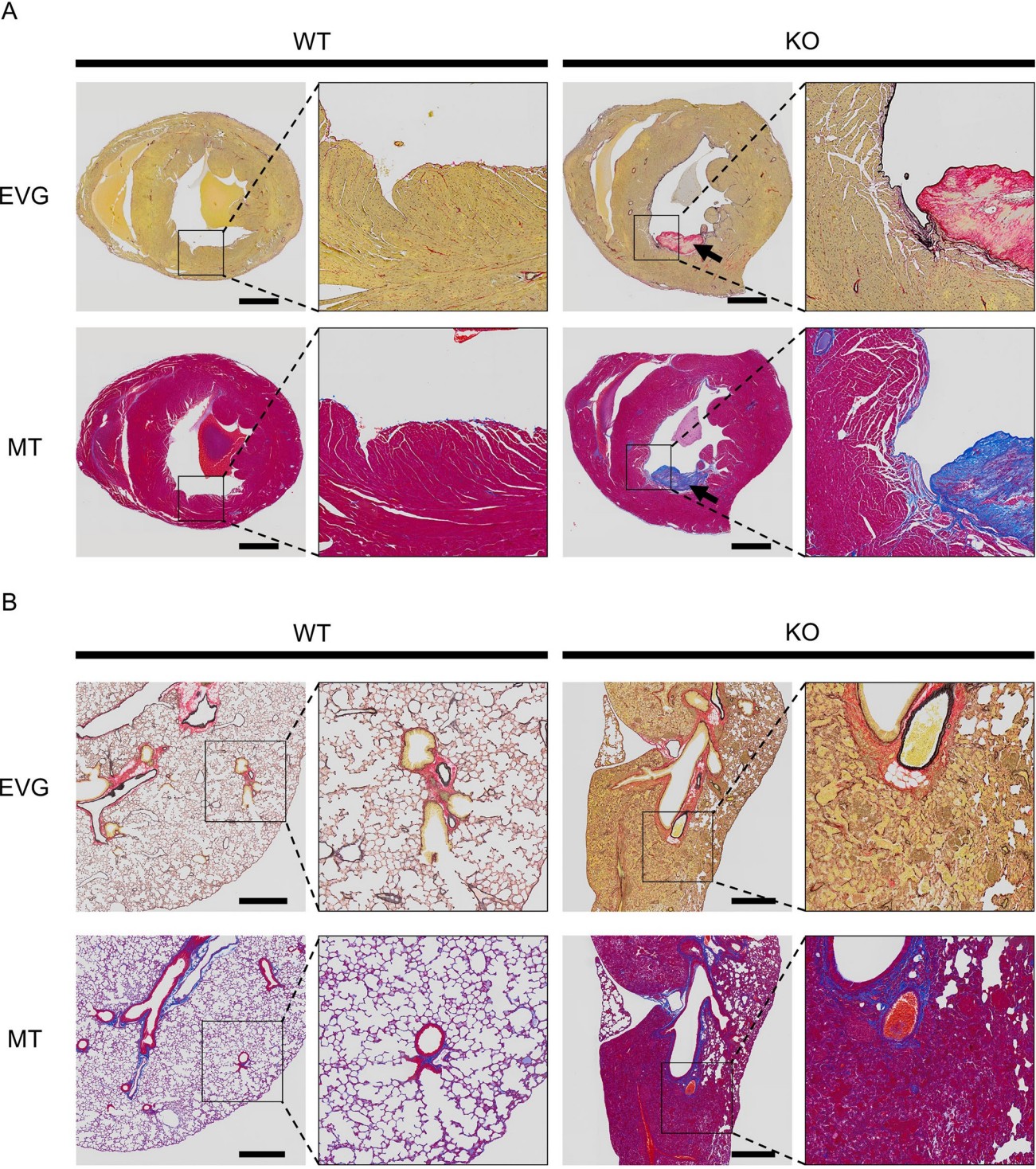

**Fig 5. Images of elastic fiber staining of hearts from WT and KO mice.** (A) Elastica van Gieson (EVG) and Masson's trichrome (MT) staining were performed using horizontally cut hearts from WT and KO mice. An enlarged view is shown on the right. Scale bars represent 1 mm. Arrow indicates the site of calcification. (B) EVG and MT staining images of lungs from WT and KO mice. An enlarged view is shown on the right. Scale bars represent 500 μm.

**Table 3. Number of fetal sacs and embryos at E11.5.**

(A)

|  | Number of fetal sacs holding fetuses of each indicated genotype at E11.5 | | |
| --- | --- | --- | --- |
|  | WT | He | KO |
| Total | 22 | 58 | 24* |
| Pregnancy Rate | 21.2% (n = 22/104) | 55.8% (n = 58/104) | 23.1% (n = 24/104) |
|  |  |  | * One case was absorbed. |
|  |  |  | χ2(2) = 2.077, n.s. |

(B)

|  | Number of fetuses of each indicated genotype at E11.5 | | |
| --- | --- | --- | --- |
|  | WT | He | KO |
| Alive | 16 | 52 | 7 |
| Dead | 6 | 6 | 17 |
| Surviving rate of fetus | 72.7% (n = 16/22) | 89.7% (n = 52/58) | 29.2% (n = 7/24) |
|  |  | χ2(2) = 30.892, p < 0.001 | |

After mating of PKCδ He mice, 11.5-day-old embryos from mothers (n = 11) were analyzed. (A) Genetic analysis of fetal sac was carried out. A Chi-squared test was applied to test whether the rate of pregnancy follows Mendelian rules. (B) The embryos with confirmed heartbeat were denoted as surviving embryos. Difference in the rate of survival among the genotypes was tested using a Chi-squared test.

Genotypes: WT, wild type mice; He, heterozygous knockout mice; KO, homozygous PKCδ-/- mice (PKCδ knockout).

differences between the PKCδ He and PKCδ WT fetuses. These findings suggest that at E11.5, PKCδ KO mice have an overall poor developmental status, and their cause of death is considered to be poor overall growth rather than organ-specific abnormalities. These results indicate that the absence of the PKCδ gene is detrimental to fetal growth.

## Discussion

In the present study, we designed a murine model lacking exon 7 of *Prckd*, which is utilized by most of the reported PKCδ splicing variants when generating PKCδ conditional knockouts. Therefore, LoxP sequences were inserted in the 5′ upstream and 3′ downstream regions of exon 7 of the *Prckd* gene encoding PKCδ, to generate PKCδ-flox mice using the C57BL/6N-derived RENKA ES cell line. By mating these mice with Cre mice according to each research objective, it is possible to produce mice with a conditional knockout of PKCδ molecules. Other studies on PKCδ knockout mice have been reported; however, they only knocked out PKCδI and δII, and did not achieve a complete knockout of all PKCδ family of proteins [17,18]. For instance, considering that the first exon of the PKCδ gene is a non-coding exon [28], Leitges M et al. generated PKCδ KO mice by eliminating the function of the second exon of the PKCδ gene. The resulting mice had a 129/SV background, developed normally, and were fertile. Moreover, these PKCδ KO mice had markedly increased arteriosclerotic lesions in the vein grafts compared with WT mice. Those mice with atherosclerotic lesions also had substantially more smooth muscle cells than the WT animals [17]. Additionally, Miyamoto et al. generated PKCδ knockout mice lacking exons 2 and 3. These mice had a 129/Sv background, were fertile, and survived up to 12 months despite the excessive proliferation of B cells and autoimmunity that was observed due to the lack of PKCδ [18]. Furthermore, in 2002, Mecklenbrauker et al. reported that PKCδ is an essential component of signaling pathways specific for inducing tolerance in B cells [19].

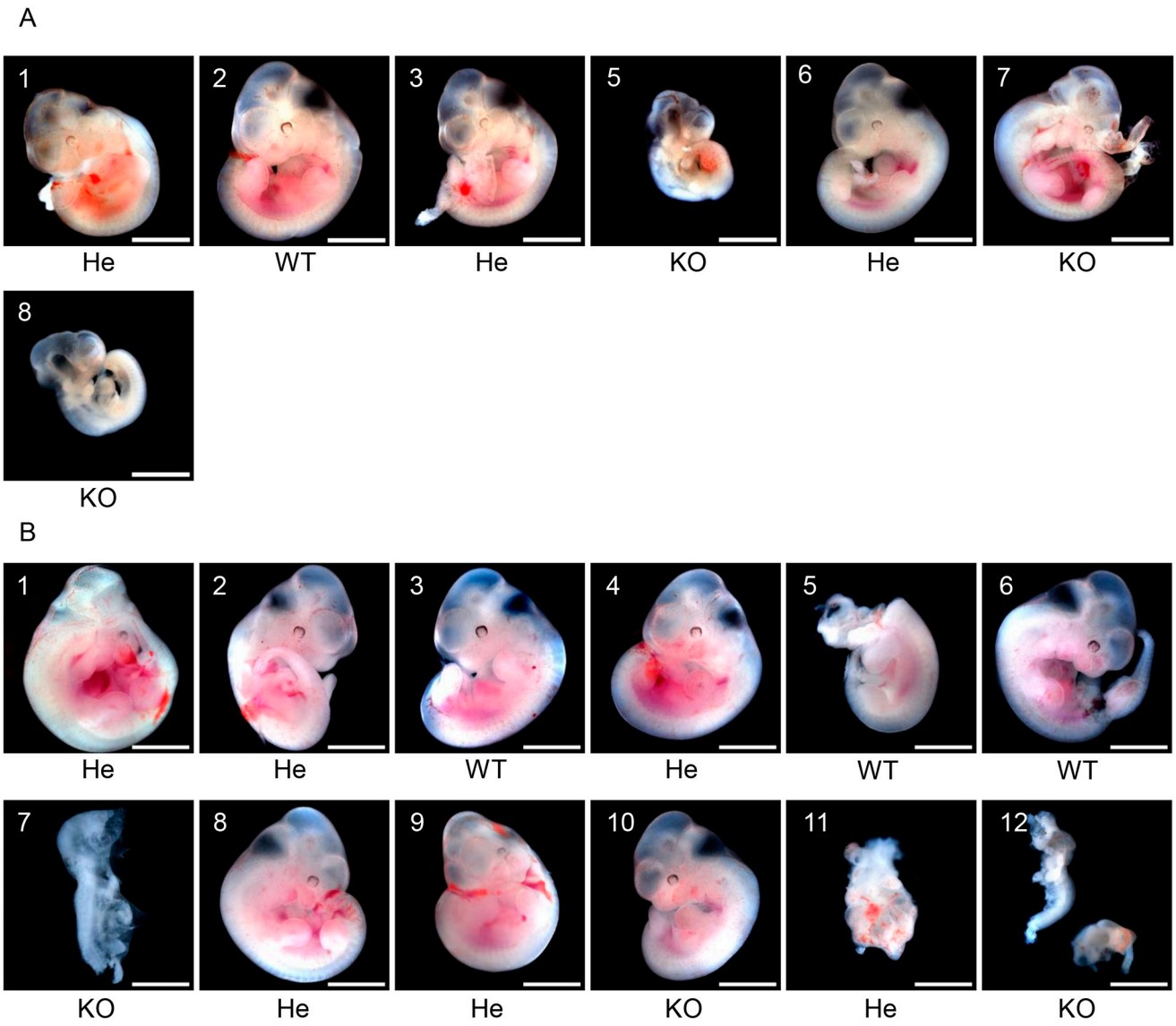

**Fig 6. 11.5-day-old embryo (E11.5) generated by crossing He × He.** The embryos of two mice out of 11 mother mice in Table 2 are shown in (A) and (B). The photographs show each individual and genotypes determined by their amniotic membranes. In (A), fetus number 4 was already absorbed. Scale bars represent 2 mm.

Considering these other related studies, ours is the first to generate PKCδ KO mice with a C57BL/6N background that can reliably delete the six PKCδ species (PKCδI, II, IV, V, VI, and VII). Many reports have described the importance of PKCδ functions, including its role in cell proliferation, cell death, and as a tumor suppressor [9–11,32–34]. In addition, the PKCδ gene produces several molecules from a single gene [20–24]. These findings suggest that a single mutation deletes more than one PKCδ molecular species, indicating the importance of this mouse model. Moreover, murine PKCδ contains a recognition sequence (DILD) for caspase-3, a member of the caspase family of cysteine proteases in its V3 domain, indicating its involvement in apoptosis [16,34–36].

Previous studies using PKCδI- and δII-null mice have investigated its role in apoptosis [37,38], defective osteoblast differentiation during embryonic development [39], defective

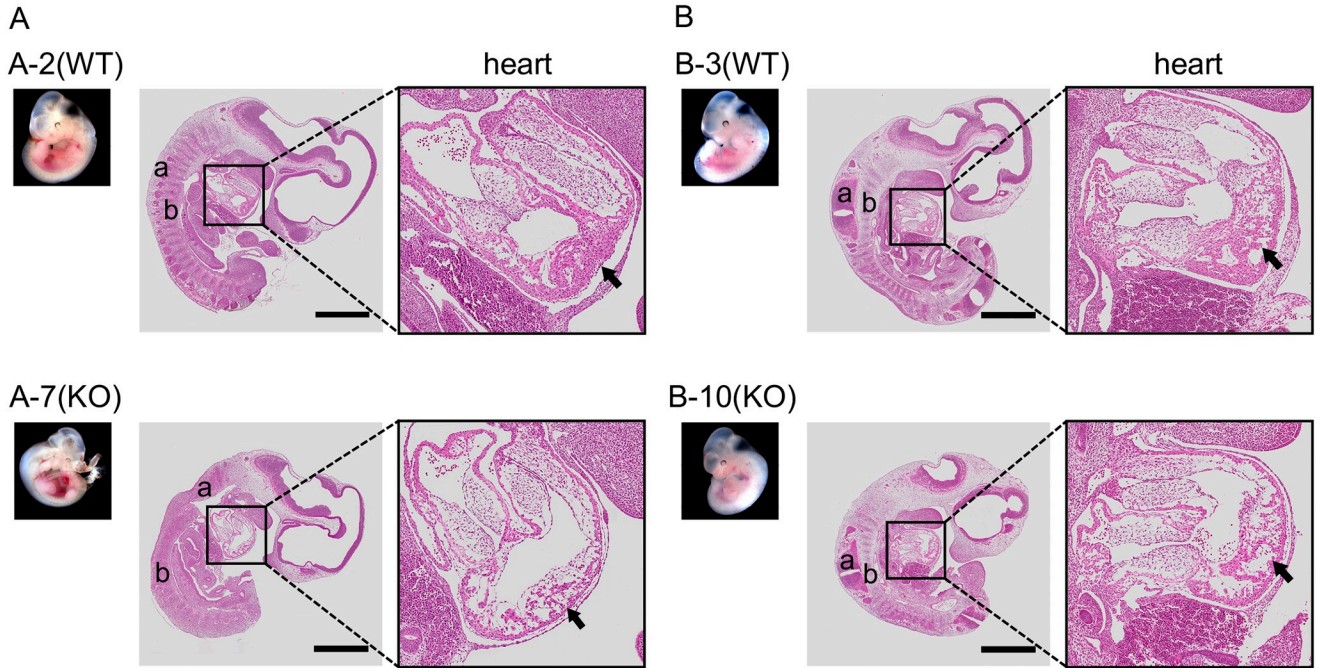

**Fig 7. Histological analysis of WT and KO mice on E11.5.** (A) Hematoxylin and eosin-stained image obtained by vertically cutting embryo numbers 2 (WT) and 7 (KO) in Fig 5A. (B) Hematoxylin and eosin-stained image obtained by vertically cutting embryos 3 (WT) and 10 (KO) in Fig 5B. Scale bars represent 1 mm. a, spinal cord; b, sclerotome.

osteoclastic bone resorption [40], as well as an increased proliferation of, and autoimmunity toward, B cells [11,18]. Sites summarizing studies on PKCδ KO mice can be found online (http://www.informatics.jax.org/marker/phenotypes/MGI:97598).

In the present study, whole-body PKCδ deficiency resulted in the survival of only 3.4% offspring, which differs significantly from previous studies, which reported no apparent issues in PKCδ KO fertility. The discrepancy in these findings may be attributed to differences in the missing PKCδ molecular species due to either differences in the gene knockout position or the mouse strain. Here we have generated a mouse that lacks exon 7. This may have resulted in the expression of a new PKCδ molecular species, resulting in a phenotypic difference from previous PKCδ KO mice. Moreover, considering that PKCδ effectively utilizes the PKCδ gene, it may be possible to discover that another splicing variant is expressed, which functions in the biological reactions of mice having exon deficiency.

It also has been reported that the experiments with sperm from PKCδ KO male mice and oocytes from WT female mice resulted in lower fertilization rates and lower early cleavage rates [41]. However, in our study, results of *in vitro* fertilization with eggs from three WT females using one male PKCδ KO mouse spermatozoa showed no differences from WT male spermatozoa when comparing fertilization rates and developmental conditions up to two cells. Together with the results of several genotypic mating combinations (Table 2), the reproductive performance of PKCδ KO males did not appear to be affected. However, due to the small number of cases, further research is expected on the reproductive ability of PKCδ KO mice.

Examination of the number of implantations due to the mating of PKCδ He showed that the numbers of WT, He, and KO fetuses were almost consistent with Mendelian law, and the implantation of PKCδ KO fertilized eggs was not low. The main reason for the birth of only 3.4% PKCδ KO mice, may be due to the developmental failure of the fetuses as observed by the

analysis of E11.5, generated by mating PKCδ He. However, PKCδ KO did not result in complete embryonic lethality. Future studies are required to determine the reason for the low PKCδ KO birth rate, including the potential use of other PKC molecules to successfully complement its function. Moreover, this low birth rate in mice may also suggest that similar issues will arise in human fetuses with PKCδ KO, given the small number of pregnancies.

PKCδ may be associated with fetal underdevelopment during pregnancy. A possible difference in the present study from previous reports on PKCδ KO mice may be the neglect of the presence of PKCδIV, V, VI, and VII produced by alternative splicing. PKCδ IV, V, VI, and VII are expressed in adult mice in a testis-specific manner and have not been studied in fetal development [21]. PKCδIV and V, and PKCδVI and VII are paired with normal and disrupted caspase-3 recognition sequences. These molecules are also deficient in the KO mice in this study. The mice generated in the present study will also be useful for further studies on PKCδIV, V, VI, and VII species.

Notably, of the nine KO mice that exhibited spontaneous death, born from crosses between PKCδ He mice, seven died within the first year of life, while all PKCδ KO mice had a short lifespan. Analysis of three 24-week-old PKCδ KO mice and one 16-week-old PKCδ KO mouse showed enlarged hearts in three animals, with two exhibiting calcification in the left ventricle or mitral valves. Moreover, staining of these heart tissues revealed an increase in elastic fibers in the endocardium, which may have led to the calcification within the heart. In addition, inflammation was observed in the lungs of three of the four PKCδ KO mice; however, an increase in fiber was not obvious. Although no calcifications were observed in the hearts of two cases, one had an enlarged heart. It was suspected that inflammation in the lungs might have resulted from a decrease in cardiac function. This is the first study to report calcification in the heart and inflammation in the lungs of PKCδ KO mice. Moreover, as many of the PKCδ KO mice died while they were still fetuses, resulting in small litter sizes, the analysis of adult animals is not considered to be sufficient. In the future, the function of PKCδ in heart and lung lesions, as well as other organs, should be investigated in a larger population.

It has been previously reported that PKCδ is expressed in the mouse heart [42]. Recently, PKCδ and PKCε double knockout mice demonstrated cardiac hypertrophy and thickening of the ventricular wall of the fetal heart [43]. However, in the present study, we observed that some mice had cardiac enlargement and calcification due to a deficiency present only in the PKCδ gene. This enlargement of the heart in PKCδ KO mice may correspond with the hypertrophy reported previously by Song et al. [43]. However, further detailed analysis is required to unambiguously ascertain the cause of the observed enlargement.

It has also been reported that PKCδ gene-deficient mice have altered immune function. PKCδ KO mice develop autoimmune diseases with aging, indicating that PKCδ plays a role in B cell tolerance of autoantigen induction [19]. Research on human lesions has advanced as well. For instance, a missense mutation (c.1528G>A) in the human PKCδ gene via a single nucleotide substitution replaces the amino acid glycine with serine (p. G510S) in children born to parents heterozygous for this missense mutation of PKCδ gene. Children homozygous for this mutation are reported to have systemic lupus erythematosus, caused by a marked decrease in PKCδ function [44]. Furthermore, patients with a homozygous missense mutation (c.1840C>T) in the human PKCδ gene, involving the substitution (p. R614W) from arginine to tryptophan, have several autoantibodies, systemic lymphadenopathy, and hepatosplenomegaly [45]. Since single mutations can cause severe symptoms, gene products must be comprehensively analyzed, particularly for genes like PKCδ, which encode multiple proteins by alternative splicing. Therefore, the mice produced in the current study have the potential to elucidate the effects of the multiple protein variants in mice, as well as humans.

PKCδ also plays a role in inflammatory diseases such as sepsis [46,47], and is reportedly associated with diabetes, with PKCδ regulating glucose production in rats [48], and increased PKCδ expression and activation in diabetic rats [49]. Moreover, continuous hyperglycemia is a common event in patients with type I and type II diabetes. It is reported that PKCδ is involved in advanced glycation end product-induced apoptosis produced by this hyperglycemic exposure [50,51]. Hence, it is expected that the mice produced in the present study will contribute to the elucidation of important functions of PKCδ and its role in various diseases.

## Supporting information

**S1 Raw images. Original gel images of Fig 2B-1 (DIG), Fig 2B-2 (EtBr) and Fig 2C.** (PDF)

## Acknowledgments

We thank the Media Technology Laboratory of Tokyo Metropolitan Institute of Medical Science for photographs and figures of mice and their tissues. We would like to thank Dr. T. Nakamachi and Dr. J. Watanabe for their cooperation in producing and maintaining PKCδ KO mice.

## Author Contributions

**Conceptualization:** Yuko S. Niino.

**Formal analysis:** Ikuo Kawashima, Yoshinobu Iguchi, Maya Yamazaki, Kenji Sakimura.

**Funding acquisition:** Yuko S. Niino, Hiroaki Kanda.

**Investigation:** Yuko S. Niino, Ikuo Kawashima, Yoshinobu Iguchi, Hiroaki Kanda, Kiyoshi Ogura, Kaoru Mita-Yoshida, Tomio Ono, Takaya Gotoh.

**Methodology:** Maya Yamazaki, Kenji Sakimura.

**Project administration:** Yuko S. Niino.

**Resources:** Satomi Yogosawa, Kiyotsugu Yoshida, Seiji Shioda.

**Supervision:** Yuko S. Niino, Kenji Sakimura, Seiji Shioda, Takaya Gotoh.

**Validation:** Yuko S. Niino.

**Visualization:** Yoshinobu Iguchi.

**Writing – original draft:** Yuko S. Niino, Ikuo Kawashima, Tomio Ono, Seiji Shioda, Takaya Gotoh.

**Writing – review & editing:** Yoshinobu Iguchi, Hiroaki Kanda, Maya Yamazaki, Kenji Sakimura, Satomi Yogosawa, Kiyotsugu Yoshida.

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
