## [Decision Letter · Decision Letter 0]

4 Jan 2021

PONE-D-20-38543

PKCδ deficiency inhibits fetal development and is associated with heart elastic fiber hyperplasia and lung inflammation in adult PKCδ knockout mice

PLOS ONE

Dear Dr. Gotoh,

Thank you for submitting your manuscript to PLOS ONE. After careful consideration, we feel that it has merit but does not fully meet PLOS ONE’s publication criteria as it currently stands. Therefore, we invite you to submit a revised version of the manuscript that addresses the points raised during the review process.

Further studies and clarifications need to be provided, including better presentation of the differences and similarities found in the new KO model relative to the old KOs, and accuracy in the description of quantitative phenotypes.

We look forward to receiving your revised manuscript.

Kind regards,

Diego Fraidenraich

Academic Editor

PLOS ONE

Journal Requirements:

Reviewers' comments:

Reviewer's Responses to Questions

**Comments to the Author**

1. Is the manuscript technically sound, and do the data support the conclusions?

Reviewer #1: Partly

Reviewer #2: Partly

Reviewer #3: Yes

2. Has the statistical analysis been performed appropriately and rigorously? 

Reviewer #1: No

Reviewer #2: Yes

Reviewer #3: Yes

3. Have the authors made all data underlying the findings in their manuscript fully available?

Reviewer #1: Yes

Reviewer #2: Yes

Reviewer #3: Yes

4. Is the manuscript presented in an intelligible fashion and written in standard English?

Reviewer #1: Yes

Reviewer #2: Yes

Reviewer #3: Yes

5. Review Comments to the Author

Reviewer #1: The authors describe a new knockout of the PKCd gene where they have targeted Exon 7. Previous knockouts have targeted the first two exons and this leaves the possibility that alternative splice forms, not using these early exons, may be expressed. In contrast, Exon 7 is required for all isoform bar IX (although this could be made clearer in Figure 1). With the new KO, embryonic survival is compromised and hardly any KO mice are generated. This is potentially important as it suggests the previous conclusion that PKCd is not needed for survival or breeding may be incorrect and it perhaps also uncovers some important in vivo roles for specific isoforms. Overall, these models could be very useful in understanding PKCd function – and this is a valid and worthwhile undertaking - but in many cases, considerable further study will be required to understand mechanism. As an initial report that PKCd may be essential, the work is quite convincing. There are however a few areas which should really be strengthened or clarified to make this case more convincingly, and there are number of other areas where current analyses could be improved.

Major Point: Clarity of the approach and comparison with other PKCd KOs

1. In the original PKCd KOs it is stated “ PKCδ–/– mice developed normally and were fertile.” https://www.jci.org/articles/view/12902 and “The PKC-δ-/- mice were viable up to 12 months of age, despite detection of auto-immune disease in these animals (see below).”https://www.nature.com/articles/416865a. The authors state in the current paper (line 75: “Although these studies reported significant findings, the PKCδ KO mice used in these studies lacked only the PKCδI and δII isoforms.” This is reastsed in the conclusion.

This is completely critical to the manuscripts originality. Both initial PKCd KO mouse models indeed targeted exons early in the gene leaving the possibility that alternative splice variants may be made. However, in Mecklenbräuker et al, it is stated that “The absence of full-length or truncated PKC-δ in lymphocytes was confirmed by immunoblot analysis of cell lysates”. Likewise in Miyamoto et al it is stated that:“Neither full-length nor truncated PKC-δ protein was detected in PKC-δ-/- mouse cells by immunoblot analysis”

While it is understood that these immunoblot analyses are far from conclusive, it would be useful to have some direct evidence that these other models do indeed still generate alternatively sliced PKCd isoforms. Are there any published studies showing alternative PKCd isoform expression in these previous KOs? Could cell lines from these KOs be tested by qPCR or Western to show other mRNAs or proteins are indeed made? The previous authors could reasonably be requested to provide cell lines to test this.

2. The authors approach was to target Exon 7 using a conditional approach, which is common to many more isoforms. However, as with the original KOs, exon specific deletions can cause all manner of unexpected effects. For example, truncated products could perhaps now be produced from the early part of the PKCd gene. Discrepancies between phenotypes might easily be caused by production of PKCd fragments acting in a dominant way. Could the authors describe why they are convinced this would not be the case? Careful use of C and N-terminal directed antibodies on lysates would be a start. qPCR to look for truncated/alternative mRNAs could be another approach.

Phenotypes of the new PKCd mouse:

There is a lack of statistical analysis throughout the manuscript. Lack of stats on all ratios being a good example. Table 1, Table 2. Chi2 tests needed on all data. In many cases this will be very convincing. For many of the embryo and tissue analyses, there are convincing differences but it is reported in an anecdotal way which could benefit from more rigorous stats.

As an example, survival data looks really convincing but it needs to be better displayed and subjected to stats. Line 236: “The lifespan of all PKCδ KO mice born …” Could this data be presented graphically as a proper survival analysis? Is there enough data fro a Kaplan Meier plot with stats to be included?

Breeding pairs from surviving PKCd KOs show convincingly that the males are fertile and provide additional indication that females may have reproductive issues. The data on the females is a little less convincing. Out of four pairs of KO x Het, three generated 3 offspring which survived to at least 4-weeks, which given the lethality of their KO is pretty good. Data with KO x WT are slightly at odds with this (all off spring died). The authors state this requires further work. This could be usefully strengthened prior to publication if numbers have been improved and also, the data might be more clearly presented with a table.

Separating causation between heart and lung pathologies is very difficult. While the authors observe changes in elastic fibres, and state this is predominant, I would urge caution on attributing this directly to loss of PKCd in this organ. With the numbers observed and the strong pathologies in both heart and lungs, further analysis is needed here with greater numbers and better statistics to (i) accurately and statistically define the problems and (ii) suggest functional experiments to test causation. This will likely involve additional conditional crosses. While I do not suggest this is done for the current report, I would suggest the text makes clear that these effects could all be secondary to other problems occurring during development.

Minor:

On line 75-77 the authors state: “PKCδ plays a critical role in B cell homeostasis and tolerance, highlighting its potential role in the treatment of autoimmune diseases [18]. “

A reference is missing from here. Two papers showed the B-cell phenotypes, both published in Nature in 2002. https://www.nature.com/articles/416860a

Line 263 “…of four KO mice, indicating bad air content, as observed by HE staining (Fig 3B). What is meant by bad air content? Do the authors mean poor lung perfusion?

Reviewer #2: Niino et al have analyzed the role of PKC delta isoforms using mouse genetics and identify requirements during heart and lung development. The scientific question, addressing PKC delta isoform diversity, and the genetic approach are excellent. However the mouse phenotyping needs to be significantly improved in order to provide any mechanistic insights into PKC delta function. A number of suggestions follow.

1. In the small percentage of surviving mice, were the authors able to confirm previously shown requirements for PKC delta in B cell and smooth muscle homeostasis?

2. The main paragraph on page 15 ending with "require further experiments" should be shortened. Can a potential maternal defect be demonstrated statistically? Can the authors speculate on a potential cause?

3. Did the authors formally demonstrated that the enlarged hearts are hypertrophied? Was cardiomyocyte cross-sectional area evaluated? Were other markers of cardiac hypertrophy evaluated? The calcification needs to be documented in more detail. Is this within ventricular muscle? Please add arrowheads to this figure to indicate the salient points. Are the authors sure the structure highlighted in Figure 4 is part of the heart rather than accumulated blood cells or debris in the ventricular lumen? The elastic fiber results should be confirmed at high magnification and with other markers. Finally, did the authors examine knockout hearts without the cardiac phenotypes?

4. The authors find that most homozygous mutant embryos die at or prior to E11.5. After histological analysis at this stage the authors conclude that cell proliferation was poor and certain organs, the heart and lung, were immature. This point needs to be analyzed in more depth. On what basis do the authors make their conclusion on proliferation? What do they mean by "immature"? Are they implying an overall growth delay? Please show how this is evident for the heart shown in B-10 compared to that in B-3 for example? The authors need to qualify the defects in the hearts in more detail, scoring for example chamber development, compact layer thickness, myocardial differentiation. Given the mortality at this time point it would also seem important to analyze earlier developmental stages.

5. Please say more about the "conspicuous deformation" seen in the lungs? What do the authors mean by the term "bad air content"?

6. Can the authors use their conditional allele to look at cell/tissue type roles of PKC delta?

7. The top panels in Figure 5 can be omitted, as can the empty panel 4. Perhaps the authors could illustrate a smaller number of control and mutant embryos at higher magnification.

Minor points

8. The pictures of the heart in Fig3 would be improved if taken under buffer (PBS) to avoid reflection.

9. Data in Table 2 should be condensed.

10. Please spell out EVG and MT on first mention in the results.

Reviewer #3: In manuscript “PKCδ deficiency inhibits fetal development and is associated with heart elastic fiber

hyperplasia and lung inflammation in adult PKCδ knockout mice“ authors Niino et al., show how depletion of PKCδ (splicing variants I, II, IV, V, VI, and VII) affects fertility and development of mice. Authors demonstrate that PKCδ -/- mice are fertile but number of PKCδ-/- offspring was reduced. Furthermore, they observed defects in heart, lung, and spleen development. Presented manuscript thus points to protentional role of PKCδ in both development and in maintenance of homeostasis. Results are clearly described. I have only minor suggestions:

- Description of Fig1 should be contain the source references

- Authors analyse four PKCδ -/- mice, which show different results. It should be summarized in table and observed phenotype should be connected to sex and age of analysed mice.

6. PLOS authors have the option to publish the peer review history of their article (what does this mean?). If published, this will include your full peer review and any attached files.

Reviewer #1: No

Reviewer #2: No

Reviewer #3: No

---

## [Author Response · Author response to Decision Letter 0]

13 Mar 2021

Response to Reviewers

Reviewer #1: The authors describe a new knockout of the PKCd gene where they have targeted Exon 7. Previous knockouts have targeted the first two exons and this leaves the possibility that alternative splice forms, not using these early exons, may be expressed. In contrast, Exon 7 is required for all isoform bar IX (although this could be made clearer in Figure 1). With the new KO, embryonic survival is compromised and hardly any KO mice are generated. This is potentially important as it suggests the previous conclusion that PKCd is not needed for survival or breeding may be incorrect and it perhaps also uncovers some important in vivo roles for specific isoforms. Overall, these models could be very useful in understanding PKCd function ? and this is a valid and worthwhile undertaking - but in many cases, considerable further study will be required to understand mechanism. As an initial report that PKCd may be essential, the work is quite convincing. There are however a few areas which should really be strengthened or clarified to make this case more convincingly, and there are number of other areas where current analyses could be improved.

Response: Thank you for your helpful feedback. In an effort to investigate the function of PKCδ, we designed PKCδ floxed mice with organ-and time-specific defects. However, unlike the results presented previously by Leitges et al., (17), Miyamoto et al., (18), and Mecklenbrauker et al., (19), PKCδ deficiency had a lethal effect on our knockout mice. Although some of the quantitative analyses are incomplete, we have chosen to publish the preliminary results in PLOS ONE, as we believe that they are critical and of high importance in the field.

The phenotypes of the previously reported PKCδKO mice have now been described in the Discussion section of the manuscript (Lines 383-393, in “Revised_Manuscript_Clean”).

Please find our point-by-point responses to each of your comments below.

Major Point: Clarity of the approach and comparison with other PKCd KOs

1. In the original PKCd KOs it is stated “ PKCδ?/? mice developed normally and were fertile.” https://www.jci.org/articles/view/12902 and “The PKC-δ-/- mice were viable up to 12 months of age, despite detection of auto-immune disease in these animals (see below).”https://www.nature.com/articles/416865a. The authors state in the current paper (line 75: “Although these studies reported significant findings, the PKCδ KO mice used in these studies lacked only the PKCδI and δII isoforms.” This is reastsed in the conclusion.

This is completely critical to the manuscripts originality. Both initial PKCd KO mouse models indeed targeted exons early in the gene leaving the possibility that alternative splice variants may be made. However, in Mecklenbrauker et al, it is stated that “The absence of full-length or truncated PKC-δ in lymphocytes was confirmed by immunoblot analysis of cell lysates”. Likewise in Miyamoto et al it is stated that:“Neither full-length nor truncated PKC-δ protein was detected in PKC-δ-/- mouse cells by immunoblot analysis”

While it is understood that these immunoblot analyses are far from conclusive, it would be useful to have some direct evidence that these other models do indeed still generate alternatively sliced PKCd isoforms. Are there any published studies showing alternative PKCd isoform expression in these previous KOs? Could cell lines from these KOs be tested by qPCR or Western to show other mRNAs or proteins are indeed made? The previous authors could reasonably be requested to provide cell lines to test this.

Response: Due to the molecular structure of the mouse PKCδ, the N-terminal antibody recognizes PKCδI, δII, and δIX; but, cannot respond to δIV, δV, δVI, and δVII. Meanwhile, the C-terminal antibody cannot recognize PKCδVI and δVII. Analysis in adult mice also revealed that PKCδIV, δV, δVI, and δVII are tissue specific. Previously, Leitges, M et al generated PKCδ KO mice, and used Santa Cruz antibodies for western blotting; however, the epitope of that antibody was not mentioned (17, J. Clin. Invest. 108:1505-1512 (2001)). Meanwhile, Mecklenbrauker, I. et al used cell lysates of PKCδ KO mice produced by Leitges, M et al to perform western blotting with anti-PKCδ antibody (BD Transduction Laboratory) and found that no full-length or truncated PKCδ was present in the lymphocytes of PKCδ null mice (19, Nature, 416, 860-865, 2002). Miyamoto et al also performed western blotting using monoclonal antibodies to mouse PKCδ (BD Transduction Laboratories) using lysates of each organ of WT mice; however, splicing variants (PKCδIV, δV, δVI, δVII, which are abundantly expressed in the testis) could not be confirmed (18, Nature, 416, 865-869, 2002). Moreover, the data in these papers consists of short cropped gel lanes, making it unclear if PKCδ species with different molecular weights were detected. If molecules of varying molecular weights are not known, it is possible that a new band may be regarded as a non-specific band or artifact due to the nature of western blotting. Since the PKCδ gene produces several molecular species, there is no antibody that can recognize all molecular species, and it is difficult to find a good antibody that can be detected by western blotting.

Moreover, we have not observed previous studies that detected PKCδ molecules with small molecular weights in KO mice. Further, unfortunately, the main investigator, YSN, has withdrawn from the study making it impossible to perform western blotting and qPCR with these KO mice.

2. The authors approach was to target Exon 7 using a conditional approach, which is common to many more isoforms. However, as with the original KOs, exon specific deletions can cause all manner of unexpected effects. For example, truncated products could perhaps now be produced from the early part of the PKCd gene. Discrepancies between phenotypes might easily be caused by production of PKCd fragments acting in a dominant way. Could the authors describe why they are convinced this would not be the case? Careful use of C and N-terminal directed antibodies on lysates would be a start. qPCR to look for truncated/alternative mRNAs could be another approach.

Response: Thank you for your comments. We designed our model to lack the exons utilized by most of the reported splice variants when creating PKCδ conditional knockout mice. As Reviewer#1 pointed out, previously reported PKCδ KO lacks exon 2 or 3 of the PKCδ gene (28, Suh KS et al. Genomics 2003, 82 57-67), with other splice variants potentially expressed. However, it was not possible to compare our mice with those previously reported. Additionally, studies on splice variants are limited and prevent us from adequately addressing the reviewer's question. We have, therefore, included this point in the revised Discussion as we are unable to fully rule out the possibility that the truncated fragment acts as a dominant negative in our KO mice (Lines 412-416, in “Revised_Manuscript_clean”). 

Phenotypes of the new PKCd mouse:

There is a lack of statistical analysis throughout the manuscript. Lack of stats on all ratios being a good example. Table 1, Table 2. Chi2 tests needed on all data. In many cases this will be very convincing. For many of the embryo and tissue analyses, there are convincing differences but it is reported in an anecdotal way which could benefit from more rigorous stats.

Response: Since the number of tables has increased, the numbers have changed.

Tables 1 and 4 were subjected to statistical analysis. In Table 1, the phenotypes of pups born by mating PKCδ He mice were statistically analyzed against Mendel's laws. In Table 4, the phenotype of 11.5-day-old embryos of mother mice (n = 11) pregnant by mating PKCδ He mice was statistically analyzed against Mendel's laws. The content described in the text are summarized in Tables 2 and 3.

As an example, survival data looks really convincing but it needs to be better displayed and subjected to stats. Line 236: “The lifespan of all PKCδ KO mice born …” Could this data be presented graphically as a proper survival analysis? Is there enough data fro a Kaplan Meier plot with stats to be included?

Response: Kaplan Meier plot with stats survivorship curves for the nine PKCδ KO mice described here by the reviewer were generated and presented in Fig.3. 

Breeding pairs from surviving PKCd KOs show convincingly that the males are fertile and provide additional indication that females may have reproductive issues. The data on the females is a little less convincing. Out of four pairs of KO x Het, three generated 3 offspring which survived to at least 4-weeks, which given the lethality of their KO is pretty good. Data with KO x WT are slightly at odds with this (all off spring died). The authors state this requires further work. This could be usefully strengthened prior to publication if numbers have been improved and also, the data might be more clearly presented with a table.

Response: This data was presented in Table 2.

It was not possible to perform multiple crosses between PKCδ KO mice and other genotypes due to the small number of KO mice and their short life span. Hence, the small number of experiments makes it difficult to obtain definitive results on parental fertility. In fact, a very large number of rearings was required to obtain data on pregnancy and birth in PKCδ KO mice, which we could not sufficiently perform. In addition, the main researcher, YSN, has retired from research, making it impossible for us to perform additional animal studies. Although PKCδ KO females may have issues completing pregnancy and delivering their pups, while the pups born from PKCδ KO females may also have problems with growth, we are unable to adequately demonstrate this due to the small numbers. We have, therefore, added in the Discussion that further experiments are needed on a larger population of mice. We believe that research on the reproductive function of PKCδ is the next step.

Separating causation between heart and lung pathologies is very difficult. While the authors observe changes in elastic fibres, and state this is predominant, I would urge caution on attributing this directly to loss of PKCd in this organ. With the numbers observed and the strong pathologies in both heart and lungs, further analysis is needed here with greater numbers and better statistics to (i) accurately and statistically define the problems and (ii) suggest functional experiments to test causation. This will likely involve additional conditional crosses. While I do not suggest this is done for the current report, I would suggest the text makes clear that these effects could all be secondary to other problems occurring during development.

Response: We believe that additional studies with a larger population size that will allow for more detailed statistical analyses, are required to assess the lesions observed in the tissues of adult PKCδ KO mice, as Reviewer #1 indicated. This has been included in the Discussion section of the revised manuscript. 

Minor:

On line 75-77 the authors state: “PKCδ plays a critical role in B cell homeostasis and tolerance, highlighting its potential role in the treatment of autoimmune diseases [18]. “

A reference is missing from here. Two papers showed the B-cell phenotypes, both published in Nature in 2002. https://www.nature.com/articles/416860a

Response: We apologize for this oversight. The study published by Dr. Mecklenbrauker, I. et al. has now been included in the manuscript (Line 79 in “Revised_Manuscript_clean”).

Line 263 “…of four KO mice, indicating bad air content, as observed by HE staining (Fig 3B). What is meant by bad air content? Do the authors mean poor lung perfusion? 

Response: As shown in Fig. 3B and Fig. 4B, it was histologically confirmed that there is an area with an poorly air space in the lungs of KO mice compared to WT.

Reviewer #2: Niino et al have analyzed the role of PKC delta isoforms using mouse genetics and identify requirements during heart and lung development. The scientific question, addressing PKC delta isoform diversity, and the genetic approach are excellent. However the mouse phenotyping needs to be significantly improved in order to provide any mechanistic insights into PKC delta function. A number of suggestions follow.

Response: Thank you very much for your careful review of our manuscript and for your helpful feedback.

In an effort to investigate the function of PKCδ, we designed PKCδ floxed mice with organ-and time-specific defects. However, unlike the results presented previously by Leitges et al., (17), Miyamoto et al., (18), and Mecklenbrauker et al., (19), PKCδ deficiency had a lethal effect on our knockout mice. Although some of the quantitative analyses are incomplete, we have chosen to publish the preliminary results in PLOS ONE, as we believe they are of high importance and critical to the field. 

The phenotypes of the previously reported PKCδ KO mice, as well as those of the PKCδ KO mice generated in the current study, are described in the revised Discussion.

Below we have addressed each of your comment in a point-by-point manner. Please note that the numbers in the figures and tables have changed.

1.In the small percentage of surviving mice, were the authors able to confirm previously shown requirements for PKC delta in B cell and smooth muscle homeostasis?

Response: The effect of PKCδ in B-cell and smooth muscle homeostasis, previously reported in studies of PKCδ KO mice, has not been confirmed by our research.

2.The main paragraph on page 15 ending with "require further experiments" should be shortened. 

Response: Thank you for your suggestion. Accordingly, we have summarized the content and shortened the paragraph.

Can a potential maternal defect be demonstrated statistically? 

Response: Unfortunately, due to the small number of KO mice and their short life span, statistical analysis of a potential PKCδ KO maternal defect could not be shown. 

Can the authors speculate on a potential cause?

Response: To date, the only analysis related to female-specific organs, including the ovaries, fallopian tube, or uterus of mice with PKCδ4-7, was an RT-PCR study performed by Kawaguchi et al on uterine tissue (20). It is, therefore, entirely speculative that PKCδ KO mouse eggs may exhibit defects in implantation through to early development in utero. Alternatively, the uterus of PKCδ KO mice may exhibit challenges with placental function or may not respond well to early fetal development in utero.

3. Did the authors formally demonstrated that the enlarged hearts are hypertrophied? 

Response: Fig. 4A shows the organs of WT and KO litter mice. Size markers have been provided. 

Was cardiomyocyte cross-sectional area evaluated? 

Response: No, cardiomyocyte cross-sectional area has not been assessed as it was difficult to align the faces of the sections, and it was not possible to analyze the cross-sectional area of tissues and cells.

Were other markers of cardiac hypertrophy evaluated? 

Response: Unfortunately, we have not analyzed cardiac hypertrophy. 

The calcification needs to be documented in more detail. Is this within ventricular muscle? 

Response: Photographs of the heart used for tissue analysis were presented in Fig 4C.

The calcified mass was not in the myocardium but in the subendocardium.

Please add arrowheads to this figure to indicate the salient points. 

Response: The arrows were added to Fig 4 and Fig 5 to show the calcified mass in the heart. 

Are the authors sure the structure highlighted in Figure 4 is part of the heart rather than accumulated blood cells or debris in the ventricular lumen? 

Response: The structures indicated by the arrows in Fig 4B and Fig 5 are white calcified masses in Fig 4C, not blood cells or debris accumulated in the ventricular lumen. 

The elastic fiber results should be confirmed at high magnification and with other markers. 

Response: We agree that identification of elastic fibers is more relevant that other markers. However, the main researcher (YSN) has retired preventing us from performing additional studies including the staining of additional markers. 

Finally, did the authors examine knockout hearts without the cardiac phenotypes?

Response: Yes, we examined the hearts of knockout mice without the cardiac phenotype. No differences from WT were observed in the hearts of knockout mice without cardiac phenotype. This has been included in the revised manuscript (Lines 291-293, in “Revised_Manuscript_clean”).

4. The authors find that most homozygous mutant embryos die at or prior to E11.5. After histological analysis at this stage the authors conclude that cell proliferation was poor and certain organs, the heart and lung, were immature. This point needs to be analyzed in more depth. On what basis do the authors make their conclusion on proliferation? What do they mean by "immature"? 

Response: Reviewer # 2 is correct. Regarding the tissue section, we thought that the cell density of the KO tissue was coarse and that the structure was simple. However, we deleted our argument on this because they lacked scientific evidence such as measurement. We are unable to perform additional analyses. Therefore, the explanation has been revised, including the deletion of the description so that the result can be confirmed only by visual inspection. 

Are they implying an overall growth delay? Please show how this is evident for the heart shown in B-10 compared to that in B-3 for example? The authors need to qualify the defects in the hearts in more detail, scoring for example chamber development, compact layer thickness, myocardial differentiation. 

Response: We agree that more in-depth analysis regarding the development of the KO heart is required. Therefore, we deleted the description of the 11.5-day embryonic heart.

Given the mortality at this time point it would also seem important to analyze earlier developmental stages.

Response: To understand the function of PKCδ in embryogenesis, we agree that it is necessary to analyze the fetal stage earlier than E11.5. We hope to address this in a future study. 

5. Please say more about the "conspicuous deformation" seen in the lungs? 

Response: I apologize for this unclear phrasing. Considering that the observations was entirely visual, I have revised the text to "deformation" and have shown this effect in Fig 4B.

What do the authors mean by the term "bad air content"?

Response: As shown in Fig. 3B and Fig. 4B, it was histologically confirmed that there is an area with an poorly air space in the lungs of KO mice compared to WT. 

6. Can the authors use their conditional allele to look at cell/tissue type roles of PKC delta? 

Response: We have created this mouse to analyze the tissue-specific functions of PKC delta. Unfortunately, since the main researcher (YSN) has retired from research, no further experimental studies are possible. In the future, we believe that this conditional knockout mouse can be used for analysis of the tissue-specific function of PKC delta. 

7. The top panels in Figure 5 can be omitted, as can the empty panel 4. Perhaps the authors could illustrate a smaller number of control and mutant embryos at higher magnification.

Response: We recreated Fig 6 as pointed out by Reviewer 2. 

Minor points

8. The pictures of the heart in Fig3 would be improved if taken under buffer (PBS) to avoid reflection.

Response: Thank you for this suggestion. However, since the mice are no longer accessible, we are unable to provide improved images. 

9. Data in Table 2 should be condensed.

Response: Table 2 has been renamed to Table 4, which presents the number of fetal sacs in each mother to demonstrate the effect on implantation of fertilized eggs. However, we condescend all individual mice into one group and presented the total number of fetal sacs to simplify the table. 

10. Please spell out EVG and MT on first mention in the results.

Response: EVG and MT were defined in the "Histological Analysis" section of the revised Results. 

Reviewer #3: In manuscript “PKCδ deficiency inhibits fetal development and is associated with heart elastic fiber

hyperplasia and lung inflammation in adult PKCδ knockout mice“ authors Niino et al., show how depletion of PKCδ (splicing variants I, II, IV, V, VI, and VII) affects fertility and development of mice. Authors demonstrate that PKCδ -/- mice are fertile but number of PKCδ-/- offspring was reduced. Furthermore, they observed defects in heart, lung, and spleen development. Presented manuscript thus points to protentional role of PKCδ in both development and in maintenance of homeostasis. Results are clearly described. I have only minor suggestions:

Response: Thank you very much for your careful review of our manuscript and for your helpful feedback.

In an effort to investigate the function of PKCδ, we designed PKCδ floxed mice with organ-and time-specific defects. However, unlike the results presented previously by Leitges et al., (17), Miyamoto et al., (18), and Mecklenbrauker et al., (19), PKCδ deficiency had a lethal effect on our knockout mice. Although some of the quantitative analyses are incomplete, we have chosen to publish the preliminary results in PLOS ONE, as we believe that they are of high importance in the field. 

The phenotypes of the previously reported PKCδ KO mice, as well as those of the PKCδ KO mice produced in the current study, are described in the revised Discussion.

Below we have addressed each of your comment in a point-by-point manner. Please note that the numbers in the figures and tables have changed.

- Description of Fig1 should be contain the source references.

Response: Thank you for pointing this out. References have been added to the Fig. 1 legend. 

- Authors analyse four PKCδ -/- mice, which show different results. It should be summarized in table and observed phenotype should be connected to sex and age of analysed mice. 

Response: The phenotypes of the four PKCδ KO mice have been summarized in Table 3.

---

## [Decision Letter · Decision Letter 1]

22 Apr 2021

PONE-D-20-38543R1

PKCδ deficiency inhibits fetal development and is associated with heart elastic fiber hyperplasia and lung inflammation in adult PKCδ knockout mice

PLOS ONE

Dear Dr. Gotoh,

Thank you for submitting your manuscript to PLOS ONE. After careful consideration, we feel that it has merit but does not fully meet PLOS ONE’s publication criteria as it currently stands. Therefore, we invite you to submit a revised version of the manuscript that addresses the points raised during the review process.

Some problems with statistics and interpretations still remain. In addition, a few minor points need to be addressed.

We look forward to receiving your revised manuscript.

Kind regards,

Diego Fraidenraich

Academic Editor

PLOS ONE

Journal Requirements:

Additional Editor Comments (if provided):

Some problems with statistics and interpretations still remain. In addition, a few minor points need to be addressed.

Reviewers' comments:

Reviewer's Responses to Questions

**Comments to the Author**

1. If the authors have adequately addressed your comments raised in a previous round of review and you feel that this manuscript is now acceptable for publication, you may indicate that here to bypass the “Comments to the Author” section, enter your conflict of interest statement in the “Confidential to Editor” section, and submit your "Accept" recommendation.

Reviewer #1: (No Response)

Reviewer #2: (No Response)

Reviewer #3: All comments have been addressed

2. Is the manuscript technically sound, and do the data support the conclusions?

Reviewer #1: Partly

Reviewer #2: Yes

Reviewer #3: Yes

3. Has the statistical analysis been performed appropriately and rigorously? 

Reviewer #1: No

Reviewer #2: Yes

Reviewer #3: Yes

4. Have the authors made all data underlying the findings in their manuscript fully available?

Reviewer #1: Yes

Reviewer #2: Yes

Reviewer #3: Yes

5. Is the manuscript presented in an intelligible fashion and written in standard English?

Reviewer #1: No

Reviewer #2: Yes

Reviewer #3: Yes

6. Review Comments to the Author

Reviewer #1: The efforts to include Kaplan Meier analysis and some statistical analysis is appreciated.

332- 339: this section is confusingly written and I think needs clarification. Are the authors saying the embryos were implanted at Mendelian ratios but that most KOs were dead at E11.5?

Table 2 still appears to lack stats. The data in this table are none-the-less interesting but the interpretation remains difficult. Het off spring die as readily as KO offspring from the KO crosses suggesting a rearing issue rather than a viability issue for the young. Drawing conclusions, clearly they KOs can produce offspring but are unable to rear the young. This is difficult to research and draw solid conclusions from which the authors acknowledge..

Reviewer #2: The manuscript has been revised and some of my earlier points answered. The following points should be addressed:

1. Lines 236-240 are not very conclusive or informative. These should be rephrased or deleted. The discussion of this point on line 424 may suffice.

2. Is there experimental justification to use the term hypertrophy on line 460 (see replies to previous comments)?

Reviewer #3: I have no other comments. All my questions and suggestions were successfully responded by authors.

7. PLOS authors have the option to publish the peer review history of their article (what does this mean?). If published, this will include your full peer review and any attached files.

Reviewer #1: No

Reviewer #2: No

Reviewer #3: No

---

## [Author Response · Author response to Decision Letter 1]

7 Jun 2021

Response to Editor Comments 

Response: We checked the reference list of our paper. No literature was retracted.

Some problems with statistics and interpretations still remain. In addition, a few minor points need to be addressed.

Response to Reviewer's Comments 

Reviewer #1: The efforts to include Kaplan Meier analysis and some statistical analysis is appreciated.

Response: We thank the reviewer for reading our revised manuscript in detail and helping us improve our paper. We have read your opinion carefully. Please find our response below. 

Comment 1: 332- 339: this section is confusingly written and I think needs clarification. Are the authors saying the embryos were implanted at Mendelian ratios but that most KOs were dead at E11.5?

Response: In lines 305-325, we described that the proportion of the number of WT, He, and KO fetal sacs at E11.5 followed Mendelian laws. However, the proportion of surviving KO embryos (with heartbeat) was significantly lower than WT and He at E11.5. We apologize if this was not conveyed appropriately. We have changed the table (Table 3 in the revised manuscript) to include data showing the number of implantations and the survival rate. Additionally, we suggested that this phenomenon might occur due to a problem with KO development in the mother's body.

Comment 2: Table 2 still appears to lack stats. The data in this table are none-the-less interesting but the interpretation remains difficult. Het off spring die as readily as KO offspring from the KO crosses suggesting a rearing issue rather than a viability issue for the young. Drawing conclusions, clearly they KOs can produce offspring but are unable to rear the young. This is difficult to research and draw solid conclusions from which the authors acknowledge.

Response: We thank the reviewer for appreciating the data from Table 2. We agree with the reviewer that these data suggest possible childcare issues. Since Table 2 does not have enough data to analyze and draw statistically significant conclusions, we have removed Table 2 and its description from the revised manuscript. 

Reviewer #2: The manuscript has been revised and some of my earlier points answered. The following points should be addressed:

Response: We thank the reviewer for reading our revised manuscript in detail and helping us improve our paper. We have read your opinion carefully. Please find our response below. 

Comment 1. Lines 236-240 are not very conclusive or informative. These should be rephrased or deleted. The discussion of this point on line 424 may suffice.

Response: We thank the reviewer for the comment. Since Table 2 did not have enough data to analyze and draw statistically significant conclusions, we have removed Table 2 and its description from the revised manuscript. 

Comment 2. Is there experimental justification to use the term hypertrophy on line 460 (see replies to previous comments)?

Response: We could not provide additional experimental data showing that the enlarged heart in a PKCδ KO mouse corresponds to cardiac hypertrophy. Therefore, we deleted the word “hypertrophy” at all relevant instances and instead rewrote all statements signifying that the heart of the PKCδ KO mouse was "enlarged". 

Reviewer #3: I have no other comments. All my questions and suggestions were successfully responded by authors.

Response: We thank the reviewer for reading our treatise in detail and for the constructive feedback.

---

## [Editor Report · Decision Letter 2]

16 Jun 2021

PKCδ deficiency inhibits fetal development and is associated with heart elastic fiber hyperplasia and lung inflammation in adult PKCδ knockout mice

PONE-D-20-38543R2

Dear Dr. Gotoh,

We’re pleased to inform you that your manuscript has been judged scientifically suitable for publication and will be formally accepted for publication once it meets all outstanding technical requirements.

Kind regards,

Diego Fraidenraich

Academic Editor

PLOS ONE
---

## [Editor Report · Acceptance letter]

23 Jun 2021

PONE-D-20-38543R2 

PKCδ deficiency inhibits fetal development and is associated with heart elastic fiber hyperplasia and lung inflammation in adult PKCδ knockout mice 

Dear Dr. Gotoh:

I'm pleased to inform you that your manuscript has been deemed suitable for publication in PLOS ONE. Congratulations! Your manuscript is now with our production department. 

Kind regards, 

on behalf of

Dr. Diego Fraidenraich 

Academic Editor

PLOS ONE